# A Magnetic Reduced Graphene Oxide Nanocomposite: Synthesis, Characterization, and Application for High-Efficiency Detoxification of Aflatoxin B_1_

**DOI:** 10.3390/toxins16010057

**Published:** 2024-01-19

**Authors:** Chushu Zhang, Haixiang Zhou, Shining Cao, Jing Chen, Chunjuan Qu, Yueyi Tang, Mian Wang, Lifei Zhu, Xiaoyue Liu, Jiancheng Zhang

**Affiliations:** 1Shandong Peanut Research Institute, Key Laboratory of Peanut Biology and Breeding (Ministry of Agriculture and Rural Affairs), Qingdao 266100, China; peanutzhangchushu@163.com (C.Z.); pro.zhouhaixiang@163.com (H.Z.); caoshining@163.com (S.C.); cjibfc@sina.com (J.C.); qu18563972291@163.com (C.Q.); yueyit@126.com (Y.T.); wmshmilyzcx@126.com (M.W.); azhulf@163.com (L.Z.); 2College of Materials Science and Engineering, Liaoning Technical University, Fuxin 125105, China; liu1226xy@163.com

**Keywords:** adsorbent, aflatoxin B_1_, detoxification, food safety, magnetic reduced graphene oxide

## Abstract

(1) Background: Safety problems associated with aflatoxin B_1_ (AFB_1_) contamination have always been a major threat to human health. Removing AFB_1_ through adsorption is considered an attractive remediation technique. (2) Methods: To produce an adsorbent with a high AFB_1_ adsorption efficiency, a magnetic reduced graphene oxide composite (Fe_3_O_4_@rGO) was synthesized using one-step hydrothermal fabrication. Then, the adsorbent was characterized using a series of techniques, such as SEM, TEM, XRD, FT-IR, VSM, and nitrogen adsorption–desorption analysis. Finally, the effects of this nanocomposite on the nutritional components of treated foods, such as vegetable oil and peanut milk, were also examined. (3) Results: The optimal synthesis conditions for Fe_3_O_4_@rGO were determined to be 200 °C for 6 h. The synthesis temperature significantly affected the adsorption properties of the prepared material due to its effect on the layered structure of graphene and the loading of Fe_3_O_4_ nanoparticles. The results of various characterizations illustrated that the surface of Fe_3_O_4_@rGO had a two-dimensional layered nanostructure with many folds and that Fe_3_O_4_ nanoparticles were distributed uniformly on the surface of the composite material. Moreover, the results of isotherm, kinetic, and thermodynamic analyses indicated that the adsorption of AFB_1_ by Fe_3_O_4_@rGO conformed to the Langmuir model, with a maximum adsorption capacity of 82.64 mg·g^−1^; the rapid and efficient adsorption of AFB_1_ occurred mainly through chemical adsorption via a spontaneous endothermic process. When applied to treat vegetable oil and peanut milk, the prepared material minimized the loss of nutrients and thus preserved food quality. (4) Conclusions: The above findings reveal a promising adsorbent, Fe_3_O_4_@rGO, with favorable properties for AFB_1_ adsorption and potential for food safety applications.

## 1. Introduction

Aflatoxins (AFs), which were first discovered in 1960, are a group of secondary metabolites of the genus *Aspergillus*, which mainly includes *Aspergillus flavus*, *Aspergillus parasiticus*, and *Aspergillus nomius* [1,2]. AFs can be harmful to humans and livestock through inhalation, ingestion, and even skin contact and exhibit cytotoxic, endocrine-disrupting, carcinogenic, and mutagenic effects [3,4]. At present, more than 20 types of AFs have been isolated and identified, among which aflatoxin B_1_ (AFB_1_) is the most common and harmful [5]. AFB_1_ impairs the intracellular membrane system, Golgi apparatus, lysosome, and mitochondrial function to affect porcine oocyte maturation quality [6]. AFB_1_ exposure decreases the total tract nutrient digestibility, nitrogen retention, total weight gain, and average daily gain of Saanen goats in a dose-dependent manner [7]. AFB_1_ pollution is widespread worldwide and poses a great threat to various industrial products, including grains and oils (especially those derived from corn and peanuts), nuts, and spices, thus leading to increasingly serious food and feed security issues [8,9]. In addition, dairy cattle can produce contaminated milk after ingesting AFB_1_-contaminated feed [10]. A recent review showed that approximately 60% to 80% of global food crops are contaminated with mycotoxins, especially AFB_1_, which contradicts the widely cited value of 25% reported by the FAO [11]. A large proportion of the population worldwide is at risk due to exposure to AFB_1_ and many other mycotoxins [8]. Due to climatic variability and environmental complexity during crop growth and grain storage, preventing AFB_1_ contamination during the cultivation and storage stages is particularly difficult, especially in rainy environments [12,13]. Therefore, the removal of AFB_1_ from contaminated agricultural products, particularly from vegetable oils because of the strong lipophilic capability of AFB_1_, has become the focus of research on food safety.

There are currently two main methods for removing AFs, namely, degradation and adsorption. To date, a number of physical, biological, and chemical methods have been employed to degrade AFB_1_ in contaminated food. Chemical detoxification, such as the application of ozone, organic acids, or strong alkaline agents, has proven to be an effective approach with relatively high efficiency. When ozone was used to degrade AFB_1_ in corn gluten meal, after a treatment time of 40 min, the mass concentration of AFB_1_ decreased from 8.621 µg·kg^−1^ to 1.93 µg·kg^−1^; however, the contents of starch and protein decreased after ozone treatment [14]. The chemicals used for detoxification may either damage the nutritional composition of the processed food or cause secondary pollution; therefore, the practical application of these chemicals is limited. Biological methods use microorganisms or enzymes to convert AFs into nontoxic metabolites. Some studies reported that after the coculture of *Aspergillus niger* and *Pleurotus ostreatus* strains with the improved production of detoxifying enzymes, the maximum AFB_1_ degradation reached 93.4% [15]; moreover, *Bacillus subtilis* JSW-1 could detoxify 62.8% of AFB_1_ in 72 h [16]. Although biological processes can degrade AFs, the long reaction time and harsh reaction conditions greatly limit their industrial application. In addition, typical physical treatments, such as irradiation and cold plasma processes, have been widely studied for the removal of mycotoxins [17]. Ghanghro et al. degraded AFs in wheat using UV light, and over 80% of AFs were removed under exposure to 254 nm shortwave UV for 160 min [18]. In another report, more than 90% of AFB_1_ was reduced via fifteen minutes of N_2_-plasma treatment at 1.5 kpps. Moreover, the toxicity to HepG_2_ cells was eliminated [19]. Physical methods are simple and inexpensive, but they are accompanied by reductions in the nutritional value of food and feed.

The other AF removal method is to use adsorbents to bind the toxins; this approach has attracted increasing amounts of attention because of its lower cost, easier operation, and better usability on a large scale. Adsorbents are added to food, where they adsorb toxins in a sponge-like fashion and are subsequently separated by filtration or centrifugation to remove the toxins [20]. Therefore, the adsorbent is critical to the toxin removal process. Currently, the most widely studied adsorbents include activated carbon, kaolinite, bentonite, and montmorillonite [21,22]. However, the application of these adsorbents is still associated with shortcomings. A large loss of nutrients inevitably occurs during the separation of small adsorbents from detoxified food after AFB_1_ adsorption. Activated carbon was recognized in one study to be an effective adsorbent, but 4.4 kg of vegetable oil was lost after treatment with one kilogram of activated carbon [23]. In comparison, some small-particle adsorbents with a larger surface area and lower mass transfer limitations have better utilization value. Graphene has the advantages of small size, large specific surface area, nontoxicity, and favorable biocompatibility [24] and is thus widely used in environmental pollutant removal, water filtration, the removal of metal ions and dyes from aqueous solutions, and other areas [25]. Furthermore, importantly, graphene has good adsorption selectivity, with the highly efficient adsorption of aromatic substances via strong π − π interactions [20,26]. Many studies have evaluated the toxicity of reduced graphene oxides (rGOs) at both the cellular and animal levels, but no toxicity has been observed [27]. Bajpai et al. studied the effect of a porous graphene aerogel on the viability of lung epithelial cells, and noncytotoxicity and biocompatibility were observed at concentrations ranging from 5–50 mg·mL^−1^ [28]. Animal experiments have shown that ultra-small nanographene can be cleared from the body after systemic administration without significant toxicity to treated mice [29]. Therefore, graphene materials are widely used in food processing, such as for the preparation of lactose-free milk [30] and food packaging materials [31]. Moreover, graphene nanoparticles with the desired surface area are suitable for AFB_1_ removal from food, especially contaminated vegetable oil. The higher adsorption efficiency and lower required usage amount of graphene nanoparticles result in less food loss compared with the other adsorbents described above. However, extreme difficulties can occur if treated liquid foods are separated from solid adsorbents of such a small size via centrifugation and filtration, which can increase the degree of inconvenience and increase energy consumption. Generally, decontamination processes need to meet several criteria to achieve industrialization: (1) effective and rapid reduction in AF content to an acceptable level, (2) no toxic residues or secondary contamination, (3) the maintenance of the nutritional value and sensory characteristics of food and feed, and (4) economic feasibility and environmental friendliness. As a result, the research and development of removal methods that satisfy all the above conditions still face significant challenges, and optimization and improvement are urgently needed [32].

Fortunately, an alternative has arisen in the form of magnetic materials, which are widely utilized owing to their easy separation when an external magnetic field is introduced [33]. Magnetic graphene not only preserves the original outstanding adsorption performance of graphene but also solves the problems of difficult separation and recycling due to the addition of magnetic nanoparticles [34]. Therefore, magnetic graphene has often been used as a new type of adsorption material to adsorb and remove pollutants in the environment, e.g., for the removal of heavy metal ions, radioactive metal ions, pesticides, and herbicides [35]. To date, there is little information related to the application of magnetic graphene nanomaterials for mycotoxin removal [36]. In addition, there are few reports regarding the optimization of the preparation conditions and the adsorption mechanism of these nanoadsorbents for mycotoxins, despite the significant influence of these factors on adsorption characteristics [9].

Our previous results showed that after successive oxidation and reduction, freshly prepared rGO had a higher adsorption capacity for AFB_1_ than graphene oxide (GO) and graphene [37]. On this basis, herein, magnetic reduced graphene oxide (Fe_3_O_4_@rGO) was synthesized via a solvothermal method to achieve the efficient adsorption and removal of AFB_1_. Then, the effects of preparation temperature and time on the adsorption ability of Fe_3_O_4_@rGO were systematically investigated, followed by the characterization of the optimal Fe_3_O_4_@rGO material through different methods and the preliminary clarification of the adsorption mechanism for AFB_1_. Furthermore, the prepared Fe_3_O_4_@rGO was applied to treat contaminated vegetable oil and peanut milk, and the effects of the adsorption process on the active ingredients were tested. The prepared Fe_3_O_4_@rGO can efficiently adsorb AFB_1_ with easy separation, no secondary pollution, and a minimal loss of nutrients, making it useful for guaranteeing food security.

## 2. Results and Discussion

### 2.1. Optimization of Fe_3_O_4_@rGO Preparation Conditions

The relatively large specific surface area of graphene endows it with desirable adsorption properties; thus, graphene is widely used in many fields and has good performance compared with other nanoadsorbents [23]. In this study, magnetic iron oxide nanoparticles were synthesized and reacted with active groups on the surface of rGO to prepare a novel magnetic composite nanoadsorbent (Fe_3_O_4_@rGO), which dramatically improved the separation of the nanomaterial.

The effects of different preparation temperatures and times on the adsorption capacity of Fe_3_O_4_@rGO for AFB_1_ were studied. Table 1 shows that different thermal reaction conditions strongly influenced the adsorption properties of the composite. Regardless of reaction time, the adsorption capacity of the material for AFB_1_ first increased and then decreased with increasing reaction temperature (180–220 °C) and reached a significant peak at 200 °C (Table 1). As shown in Table 1, there was no significant difference in adsorption capacity at reaction times of 6, 8, or 10 h (200 °C). As a result, temperature seemed to be a more significant influencing factor. As displayed in Table 1, the AFB_1_ adsorption capacity of the composite prepared at 200 °C for 6 h was the highest, reaching 50.25 ± 0.71 mg·g^−1^ at an initial AFB_1_ concentration of 60 mg·L^−1^.

### 2.2. Characterization of Fe_3_O_4_@rGO

To obtain further insight into the various properties of the newly prepared Fe_3_O_4_@rGO, several methods, such as transmission electron microscopy (TEM), scanning electron microscopy (SEM), X-ray powder diffraction (XRD), Fourier transform infrared (FTIR) spectroscopy, and vibrating sample magnetometry (VSM), were used to comprehensively characterize the magnetic nanoadsorbent.

SEM and TEM images of the samples prepared at different temperatures are shown in Figure 1 (A and B, C and D, and E and F correspond to the TEM and SEM images of the materials synthesized at 180, 200, and 220 °C), illustrating that the Fe_3_O_4_@rGO samples had 2D structures with a nanometer-scale longitudinal size and micron-scale transverse dimension [38,39]. Although layered morphologies formed under all three temperatures, as observed in the TEM images (Figure 1A,C,E), similar to the findings of many previous reports [40], the material synthesized at 180 °C seemed to be thicker with a greater number of laminar structures than that synthesized at 200 °C (Figure 1A,C). In addition, in the SEM images, a smoother surface with fewer wrinkled structures was observed for the rGO sheets produced at 200 °C (Figure 1B,D,F). In addition, after preparation of the rGO nanomaterials, Fe_3_O_4_ particles with a uniform size of a few tens of nanometers were distributed on the surface of the rGO sheets (Figure 1). However, when 220 °C was used, too many spherical Fe_3_O_4_ particles extensively agglomerated on the surface of rGO, which occupied many sites for AFB_1_ adsorption (Figure 1E,F). Therefore, the preparation temperature can strongly affect the spatial structure of graphene and loading of Fe_3_O_4_ nanoparticles, and excessively thick rGO sheets and too high a loading of Fe_3_O_4_ particles are unfavorable for the adsorption capacity of Fe_3_O_4_@rGO.

Fe_3_O_4_@rGO samples prepared at different reaction temperatures for the same time period (6 h) were studied via XRD analysis. As shown in Figure 2, the characteristic peak located at 24.6° indicates that graphene was present in the composites [41]. Furthermore, six other typical diffraction peaks appear (2θ = 30.3°, 35.6°, 43.5°, 53.8°, 57.4°, and 62.8°), corresponding to the (220), (311), (400), (422), (511), and (440) lattice planes, respectively [36,42]; the (311) peak was much more intense than the other five peaks (Figure 2), possibly due to the predominant orientation of the (311) plane [36]. This XRD pattern confirmed the presence of crystalline Fe_3_O_4_ with a face-centered cubic structure at the nanoscale based on JCPDS Standard Cubic Fe_3_O_4_ XRD pattern No. 85–1436 [34], which greatly facilitated the separation and recycling of the nanocomposite adsorbent. Moreover, as shown in Figure 2, with increasing preparation temperature, the characteristic Fe_3_O_4_ crystal peaks gradually increased, indicating that temperature affects the loading of Fe_3_O_4_, and the higher the temperature was, the more loaded Fe_3_O_4_ particles there were. These findings are consistent with the above TEM results.

Figure 3 shows the FT-IR absorption spectra of the Fe_3_O_4_@rGO samples. The results revealed two predominant adsorption peaks at approximately 1580 cm^−1^ and 547 cm^−1^. The former characteristic peak could be attributed to C=C stretching of the sp^2^ carbon skeleton network of graphene [1], and the latter was related to the tensile vibration of the Fe–O bond in Fe_3_O_4_ [43], which demonstrated that the Fe_3_O_4_ nanoparticles were supported on the graphene material. Interestingly, several spectral signals located between 1750 cm^−1^ and 1500 cm^−1^ could be observed despite having weak intensities (Figure 3); these peaks were probably due to the active functional groups generated during oxidation and reduction, such as carbonyl C=O (1725 cm^−1^), carboxyl O=C–O (1380 cm^−1^), and alkoxy C–O (1062 cm^−1^) [36].

The nitrogen adsorption–desorption results for Fe_3_O_4_@rGO produced at 200 °C are shown in Figure 4. The isotherm of this material could be classified as Langmuir IV type with an H3-type hysteresis loop and a capillary condensation step (Figure 4A), which are common in laminar materials, indicating that this magnetic composite has a mesoporous structure with fine pore connectivity but a relatively non-uniform pore size distribution (Figure 4B) [44]. The specific surface area of the Fe_3_O_4_@rGO produced at 200 °C for 6 h was estimated to be 87.094 m^2^·g^−1^ following the Brunauer–Emmett–Teller (BET) method, while its average pore diameter and pore volume were estimated to be 3.428 nm and 0.483 cm^3^·g^−1^, respectively, based on the Barrett–Joyner–Halenda (BJH) model. These values were greater than those of the materials prepared at 180 and 220 °C (Appendix A). A greater surface area and increased pore volume indicate more exposed functional groups, which further lead to a greater adsorption capacity for AFB_1_ (Table 1).

The typical magnetic hysteresis curve of Fe_3_O_4_@rGO prepared at 200 °C is shown in Figure 5. No hysteresis was evident, and the residual magnetization and coercivity were approximately zero (Figure 5), which suggested that Fe_3_O_4_@rGO is a typical superparamagnetic material [36]. The saturation magnetization of Fe_3_O_4_@rGO was 33.55 emu/g at room temperature, indicating that Fe_3_O_4_@rGO has a strong magnetic response to magnetic separation. This prediction was confirmed using the magnetic separation test, as exhibited by the inset in Figure 5. The adsorbent could be completely and quickly separated under an external magnetic field, and the remaining clear supernatant indicates that simple and efficient solid–liquid separation could be easily achieved by using only a permanent magnet.

### 2.3. Effects of pH and Temperature on the Adsorption Capacity of Fe_3_O_4_@rGO

To assess the adsorption capacity of Fe_3_O_4_@rGO for AFB_1_, the pH and temperature were optimized. As shown in Figure 6A, at pH 3–5, the adsorption capacity of AFB_1_ increased with increasing pH; however, changes in pH had little effect on the adsorption capacity at pH 5–7, and an increase in adsorption capacity did not occur until reaching pH 8. Acidic conditions lead to the protonation of adsorbent surfaces and adsorbate molecules, resulting in repulsive forces that hinder the adsorption of AFB_1_; therefore, an increase in pH leads to deprotonation and promotes attraction between the adsorbent and the adsorbate molecules to some extent [45]. Electrostatic attraction, hydrogen bonding, and π−π interactions among AFB_1_ molecules and the Fe*_3_*O*_4_*@rGO surface could also occur during the adsorption process [24]. In this experiment, the unadjusted pH was 6–7; most foods, such as milks, oils, and juice beverages, are often neutral or slightly acidic [46], falling in the optimal pH range for adsorption by Fe_3_O_4_@rGO; therefore, the experiment could be carried out at natural pH without any adjustment. Moreover, the optimal temperature was investigated. As presented in Figure 6B, the adsorption capacity was positively correlated with temperature and reached a maximum at 45 °C. Although within a certain range, a higher temperature may improve mass transfer and reduce the viscosity of the treated sample, a decreasing trend in adsorption capacity was found at temperatures higher than 45 °C (Figure 6B). However, little difference in performance was observed at 25–45 °C, indicating that adsorption by Fe_3_O_4_@rGO could proceed at room temperature without heating, which simplifies the operation and is conducive to reducing energy consumption and production costs.

### 2.4. AFB_1_ Adsorption Isotherm

In this section, AFB_1_ adsorption was carried out continuously using a series of initial AFB_1_ concentrations (0.02–100 mg·L^−1^) and a 1 mg·mL^−1^ adsorbent dose under an unadjusted pH and at constant temperatures of 25, 35, and 45 °C. Once the AFB_1_ concentration reached equilibrium, the adsorption isotherm was plotted (Figure 7) according to the adsorption capacity (*Q*_e_, mg·g^−1^) as a function of the AFB_1_ equilibrium concentration (Figure 7).

The adsorption isotherm data were analyzed via the Langmuir model, Freundlich model, and Temkin model to estimate the adsorption characteristics, saturated adsorption capacity and surface properties of Fe_3_O_4_@rGO in the liquid system [47]. The general Langmuir, Freundlich, and Temkin isotherm models are presented in Equations (1)–(3), respectively.
(1)$$\frac{C_{e}}{Q_{e}}=\frac{C_{e}}{Q_{max}}+\frac{1}{{K_{L}Q}_{max}}$$
(2)$$lnQ_{e}=lnK_{F}+\frac{1}{n}lnC_{e}$$
(3)$${\;\;\;\;\;\;\;Q}_{e}=B_{T}lnC_{e}+B_{T}lnA_{T}$$

In the equations above, *C*_e_ is the AFB_1_ content in solution at equilibrium (mg·L^−1^); *Q*_e_ (mg·g^−1^) is the adsorption capacity; *Q*_max_ (mg·g^−1^) is the maximum adsorption capacity; *K*_L_ represents the Langmuir adsorption constant (L·mg^−1^), which is related to the adsorption free energy; *K*_F_ (mg·g^−1^) and *n* are Freundlich constants that represent the adsorption capacity and adsorption strength, respectively; *A*_T_ (L·mol^−1^) is the equilibrium bonding constant related to the maximum bonding energy; and *B*_T_ (J·mol^−1^) is a constant related to the adsorption heat.

Table 2 shows that the Langmuir model provided the best fit to the adsorption data among the three isotherm equations, indicating that the adsorption of AFB_1_ on Fe_3_O_4_@rGO was achieved via monolayer formation (*R*^2^ > 0.99) [9]. The calculated Langmuir *Q*_max_ values for AFB_1_ were 76.34, 78.74, and 82.86 mg·g^−1^ at 25, 35, and 45 °C, respectively. Compared with the other reported adsorbents displayed in Appendix A, Fe_3_O_4_@rGO exhibited good adsorption performance towards AFB_1_ [21,22,36,44,48,49,50,51,52].

### 2.5. Adsorption Kinetics of AFB_1_ by Fe_3_O_4_@rGO Composites

Adsorption kinetics experiments were carried out at different initial AFB_1_ concentrations (20, 40, and 60 mg·L^−1^) and various temperatures (25, 35, and 45 °C) for a range of contact times (Figure 8). The adsorption efficiency of the material increased sharply in the first 30 min under all the conditions (Figure 8), indicating that the prepared material exhibited rapid and efficient adsorption of AFB_1_ because of the abundant empty adsorption sites [53,54]. In addition, the above results suggested that a higher temperature in the range of 25–45 °C was beneficial for adsorption (Figure 6B), which was further confirmed in this section. For all three initial AFB_1_ concentrations, adsorption reached equilibrium most quickly at 45 °C, while it took the most time at 25 °C (Figure 8). With an initial AFB_1_ concentration of 60 mg·L^−1^, the adsorption capacity at 5 min reached 27.02 ± 0.36, 27.78 ± 0.69, and 30.78 ± 0.17 mg·g^−1^ at 25 °C, 35 °C, and 45 °C, respectively (Figure 8C). At the beginning of adsorption, a higher adsorbate concentration provides a strong driving force for rapid adsorption by adsorbent molecules [39]. Therefore, as shown in Figure 8, at the same temperature, a higher AFB_1_ concentration promoted faster adsorption.

The pseudo-first-order (PFO) and pseudo-second-order (PSO) kinetic models were used to analyze the kinetic data by using Equations (4) and (5), respectively, to determine the mechanism of adsorption [55,56]. k_1_ (min^−1^) is the PFO rate constant; Q_e_ (mg·g^−1^) and Q_t_ (mg·g^−1^) are the equilibrium adsorption amount and the adsorption quantity at time t, respectively; and k_2_ (g·mg^−1^·min^−1^) is the PSO rate constant.
(4)$$ln\left(Q_{e}-Q_{t}\right)=lnQ_{e}-K_{1}t$$
(5)$$\frac{t}{Q_{t}}=\frac{1}{K_{2}Q_{e}^{2}}+\frac{t}{Q_{e}}$$

The corresponding parameters for both the kinetic models and the correlation coefficients (*R*^2^) are shown in Table 3. The *R*^2^ of the PSO model (0.9858–1.0000) was higher than that of the PFO model (0.7256–0.9458), and the theoretical *Q*_e,cal_ calculated using the PSO model approached the experimental *Q*_e,exp_. These results suggested that AFB_1_ adsorption by Fe_3_O_4_@rGO closely conformed to the PSO model, further suggesting that the adsorption process mainly involved chemical adsorption. AFB_1_ molecules may form chemical bonds with one another through the allocation or exchange of electrons between the hydrophilic sites of Fe_3_O_4_@rGO and AFB_1_ [57].

### 2.6. Thermodynamics of AFB_1_ Adsorption by Fe_3_O_4_@rGO

In addition, to further explore the adsorption mechanism, the influence of temperature on the adsorption process was investigated by measuring the thermodynamic constants of Fe_3_O_4_@rGO adsorption of AFB_1_. The thermodynamic constants of adsorption, including the Gibbs free energy (Δ*G*^0^), standard entropy (Δ*S*^0^), and standard enthalpy (Δ*H*^0^), were calculated using the van ‘t Hoff equations displayed in Equations (6)–(8) [58,59], where *K*c is the thermodynamic equilibrium constant, R is the ideal gas constant (8.3145 J·mol^−1^·K^−1^), and *T* is temperature (°C).
(6)$$K_{c}=\frac{Q_{e}}{C_{e}}$$
(7)$$\ln\left(K_{c}\right)=-\frac{∆H{}^{\circ}}{RT}+\frac{∆S{}^{\circ}}{R}$$
(8)∆G°=∆H°−T∆S°

Table 4 lists the values of the thermodynamic parameters. The positive Δ*H*^0^ (29.41–59.49 kJ·mol^−1^) and negative Δ*G*^0^ (−12.86 to −7.49 kJ·mol^−1^) indicated that the adsorption process of the material was spontaneous and endothermic [59]. Δ*H*^0^ values for chemisorption are greater than 40 kJ·mol^−1^, while Δ*H*^0^ values for physical adsorption are usually less than 20 kJ·mol^−1^ [9,60]. These results indicated that both physical and chemical adsorption occurred in the Fe_3_O_4_@rGO–AFB_1_ interaction system, but the latter played the major role in the adsorption process, especially under high concentrations of adsorbate, corresponding to the above adsorption kinetics results.

### 2.7. Quality of Treated Foods

The current study presents a potent composite nanomaterial adsorbent, Fe_3_O_4_@rGO, with good potential for industrial application in AFB_1_ detoxification. However, the retention of nutrients after the detoxification process is another very meaningful factor for assessing this potential adsorbent. To examine nutrient retention, the contents of several essential nutritional components were detected after the use of this magnetic composite adsorbent for adsorption. According to the above results (Figure 6B), the adsorption performance of Fe_3_O_4_@rGO at 25 °C for 1 h was comparable to that at other trial temperatures; notably, nutritional components may be lost on a larger scale at much higher temperatures. Therefore, room temperature was selected for the adsorption process, which not only is conducive to nutrient retention but also helps reduce the production cost.

In this section, activated carbon was used for comparison to study the effects of different amounts of adsorbent materials on nutrients in vegetable oil and peanut milk, and the results are displayed in Table 5. When the dosage of adsorbent was 4 mg/mL, the removal rate of AFB_1_ by Fe_3_O_4_@rGO exceeded 90%. In contrast, even when the dosage of activated carbon reached 160 mg/mL, only a 55.63% removal rate was achieved, which cannot satisfy industrial demand. Additionally, the loss rate of both the treated foods after the adsorption process with activated carbon was much greater than that with Fe_3_O_4_@rGO. Although Fe_3_O_4_@rGO inevitably adsorbed several key nutritional components, such as oryzanol and phosphatide, the loss of these compounds was less than that observed with activated carbon application. The material prepared in this study can minimize the loss of nutrients and retain active substances, especially proanthocyanidins and resveratrol, in peanut milk, thus maintaining the quality of the treated foods.

## 3. Conclusions

In this study, a high-performance Fe_3_O_4_@rGO nanoparticle composite for AFB_1_ adsorption was prepared using a one-step solvothermal method. The optimal Fe_3_O_4_@rGO was obtained at 200 °C with a reaction time of 6 h. The preparation temperature significantly affected the structure and adsorption performance of Fe_3_O_4_@rGO. TEM and SEM images showed that the material had a 2D structure with a nanoscale longitudinal dimension, and spherical particles of Fe_3_O_4_ were uniformly distributed on the graphene sheets. On the basis of other characterizations, the high specific surface area and pore volume of Fe_3_O_4_@rGO and its abundant exposed functional groups contributed to its strong adsorption capacity for AFB_1_. In addition, the results of isotherm, kinetic, and thermodynamic analyses of AFB_1_ adsorption indicated that Fe_3_O_4_@rGO, with a maximum adsorption capacity of 82.64 mg·g^−1^ at 45 °C, rapidly and efficiently adsorbed AFB_1_ under proper heating conditions mainly through chemical adsorption via a spontaneous endothermic process. Furthermore, the current nanomaterial can minimize the loss of nutrients and maintain the quality of treated foods. This study provides new insight into the preparation of magnetic composite adsorbents and presents a potential candidate for use in the food industry; this material is safe and environmentally friendly, can highly effectively detoxify AFB_1_, and causes little nutritional loss. Future work will focus on the application of Fe_3_O_4_@rGO for the adsorption of diverse mycotoxins, including but not limited to AFB_1_.

## 4. Materials and Methods

### 4.1. Materials

An AFB_1_ standard was obtained from Beijing Huaan Maike Biotechnology Co., Ltd. (Beijing, China), and was stored in the dark at 4 °C. Standards of oryzanol, proanthocyanidins, and resveratrol were purchased from the Sigma-Aldrich Company (St. Louis, MO, USA); other solvents and chemicals were of analytical grade and were obtained from commercial sources.

### 4.2. Preparation of Fe_3_O_4_@rGO

GO was prepared by an improved Hummers method using graphite powder [61], and then, Fe_3_O_4_@rGO was prepared via one-step hydrothermal fabrication. Briefly, 400 mg of GO was added to 60 mL of ethylene glycol, 0.65 g of anhydrous iron chloride and 2.6 g of anhydrous sodium acetate were added, and the mixture was stirred for more than 0.5 h. The mixture reacted at different temperatures for several hours in a reactor. A black magnetic material was obtained after cooling to room temperature and washing with anhydrous ethanol to clarify the supernatant. Thereafter, the black precipitate was freeze-dried. Finally, Fe_3_O_4_@rGO was obtained by grinding the dried black particles with a glass mortar.

### 4.3. Optimization of Fe_3_O_4_@rGO Preparation Conditions

The effects of different reaction temperatures (180, 200, and 220 °C) and times (4, 6, 8, and 10 h) on Fe_3_O_4_@rGO preparation were investigated based on the adsorption capacity (mg·g^−1^) of the prepared Fe_3_O_4_@rGO for AFB_1_. The adsorption experiments of AFB_1_ in aqueous solution were carried out by using Fe_3_O_4_@rGO with a shaking rate of 150 rpm. One milligram of AFB_1_ powder was dissolved in 1 mL of methanol to prepare an AFB_1_ methanol solution, which was subsequently diluted with deionized water. One milligram of the as-prepared material was added to 1 mL of AFB_1_ solution at an original concentration of 60 mg·L^−1^, and adsorption was carried out at 25 °C for 12 h. Afterwards, the Fe_3_O_4_@rGO was separated using a magnetic field, and the supernatant was collected to measure the residual amount of AFB_1_ by ELISA [62]. The AFB_1_ adsorption quantity of the materials was calculated using Equation (9):(9)$$Q=\frac{C_{0}-C_{e}}{m}V$$

*C*_0_ (mg·L^−1^) and *C*_e_ (mg·L^−1^) represent the concentrations of AFB_1_ in aqueous solution at the beginning and after the adsorption reaction, respectively; *V* (L) is the volume of the AFB_1_ solution; *m* (g) is the adsorbent dosage; and *Q* (mg·g^−1^) is the adsorption capacity for AFB_1_.

### 4.4. Characterization of Fe_3_O_4_@rGO

The morphology of the samples was observed with a TESCAN MIRALMS field emission scanning electron microscopy (SEM) instrument, and the sizes of the as-prepared products were examined on a JEOL JEM-1200EX transmission electron microscopy (TEM) instrument. Fourier transform infrared (FT-IR) spectra were obtained with a Thermo Scientific Nicolet iS20 spectrometer in the spectral range of 4000–500 cm^−1^ with a resolution of 1 cm^−1^. The crystal structures of the samples were characterized via X-ray diffraction (XRD) with a Rigaku Ultima IV instrument manufactured by Rigaku Ultima IV of Japan. The scanning wavelength, voltage, and current were set to 0.1542 nm, 40 kV, and 40 mA, respectively. The scanning angle was 5–90°. The sample analysis results were compared with the ICDD-PDF standard card to determine the phase. Vibrating sample magnetometry (VSM; LakeShore7404) was conducted using a Lakeshore vibrating sample magnetometer at room temperature. Nitrogen sorption–desorption experiments were performed on a Micromeritics ASAP 2460 instrument at −196 °C. The specific surface area and pore size distribution were calculated with the Brunauer–Emmett–Teller (BET) method and the Barrett–Joyner–Halenda (BJH) model, respectively.

### 4.5. AFB_1_ Adsorption Experiments

In this section, the influences of various conditions on the adsorption capacity of Fe_3_O_4_@rGO for AFB_1_ were investigated. First, the optimum pH and temperature were measured with a 1 mg·mL^−1^ adsorbent dose and 40 mg·L^−1^ AFB_1_ at different pH values (3–8) and various temperatures (20–50 °C) with shaking at 150 rpm for 1 h. Then, in the adsorption isotherm study, the AFB_1_ adsorption capacity (mg·g^−1^) was assayed based on the optimal adsorption conditions and a series of AFB_1_ concentrations (0.02–100 mg·L^−1^). Thereafter, in the adsorption kinetics study, the adsorption amounts were determined at diverse time intervals under given temperatures and AFB_1_ concentrations. Finally, an adsorption thermodynamics study was performed. For each adsorption experiment, a control group without adsorbent was used as the blank group. All the experiments in this study were conducted in triplicate.

### 4.6. Analysis of Active Substances

The determination of oryzanol was carried out using high-performance liquid chromatography (HPLC) following the method of Rashid et al. [63]. The phosphatide content was determined via the molybdenum blue colorimetric method according to the National Standard of the People’s Republic of China (GB/T5537-2008). In brief, phosphatides were burned to form phosphorus pentoxide, which was then converted into phosphoric acid using hot hydrochloric acid. After the phosphoric acid was exposed to sodium molybdate, it formed sodium phosphomolybdate, which was subsequently reduced to molybdenum blue via hydrazine sulfate. The absorbance of molybdenum blue was measured at a wavelength of 650 nm using a spectrophotometer [64]. Proanthocyanidins were detected using a spectrophotometric method (UV-2700, Kyoto, Japan) as described previously [65], and the resveratrol content was determined through HPLC [66].

### 4.7. Statistical Analysis

All the measurements were conducted in triplicate, and the results are expressed as the mean ± standard deviation (SD). Analysis of variance (ANOVA) and Duncan’s multiple comparison were performed by using the SPSS 17.0 package to determine the significant differences between groups. *p* values < 0.01 were considered to be extremely significant.

## Figures and Tables

**Figure 1 toxins-16-00057-f001:**
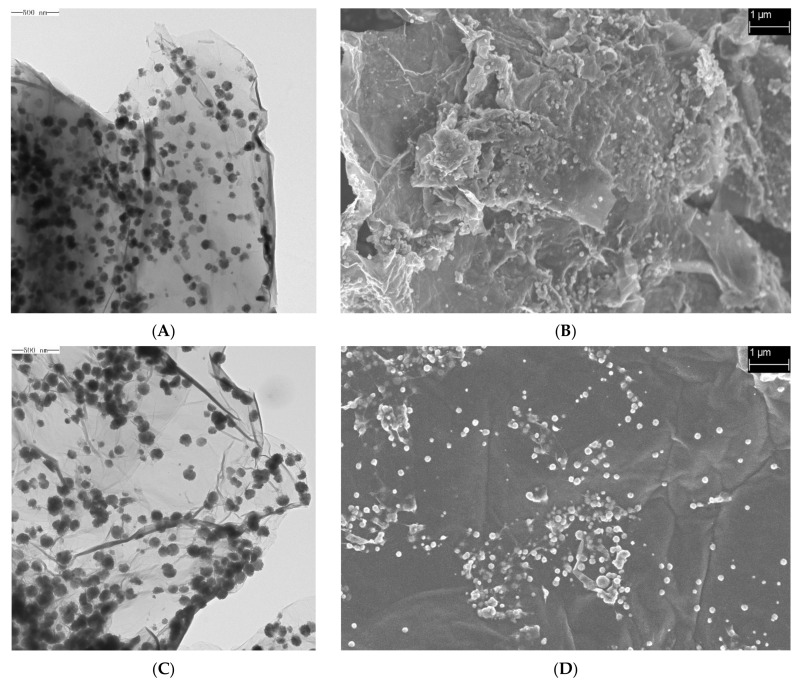
TEM and SEM images of Fe_3_O_4_@rGO prepared at different reaction temperatures. (**A**) TEM image of Fe_3_O_4_@rGO prepared at 180 °C for 6 h; (**B**) SEM image of Fe_3_O_4_@rGO prepared at 180 °C for 6 h; (**C**) TEM image of Fe_3_O_4_@rGO prepared at 200 °C for 6 h; (**D**) SEM image of Fe_3_O_4_@rGO prepared at 200 °C for 6 h; (**E**) TEM image of Fe_3_O_4_@rGO prepared at 220 °C for 6 h; (**F**) SEM image of Fe_3_O_4_@rGO prepared at 220 °C for 6 h.

**Figure 2 toxins-16-00057-f002:**
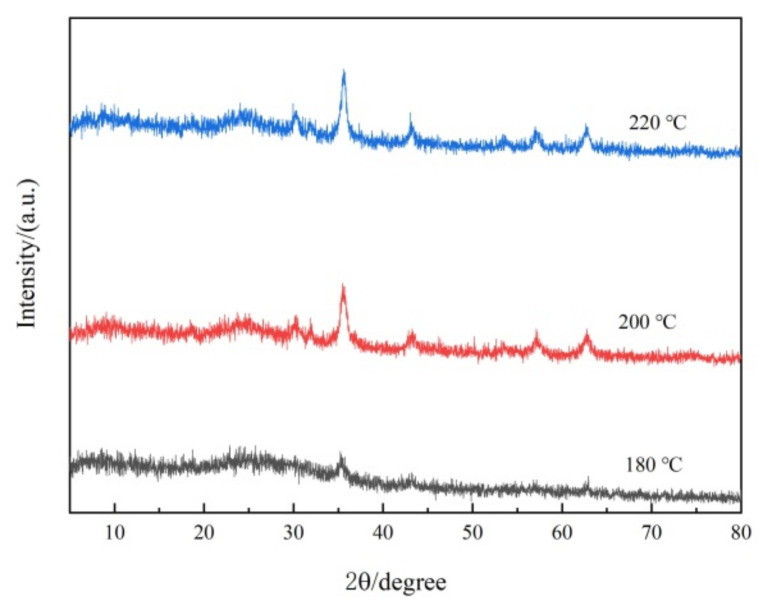
XRD patterns of Fe_3_O_4_@rGO samples prepared at different temperatures for 6 h.

**Figure 3 toxins-16-00057-f003:**
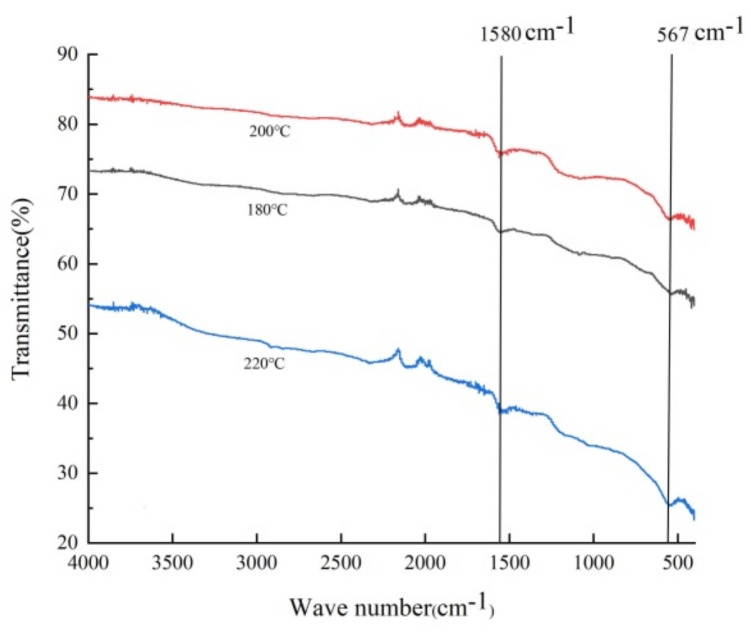
FT-IR spectra of Fe_3_O_4_@rGO prepared at different temperatures (180 °C, 200 °C, 220 °C) for 6 h.

**Figure 4 toxins-16-00057-f004:**
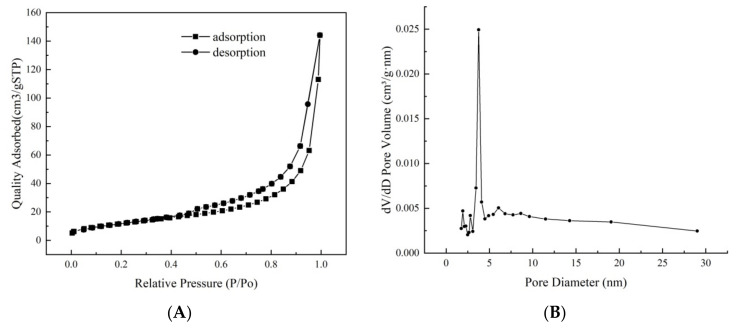
Nitrogen adsorption–desorption isotherm and pore size distribution profile of Fe_3_O_4_@rGO. (**A**) Nitrogen adsorption–desorption isotherm of Fe_3_O_4_@rGO prepared at 200 °C for 6 h; (**B**) pore size distribution profile of Fe_3_O_4_@rGO prepared at 200 °C for 6 h.

**Figure 5 toxins-16-00057-f005:**
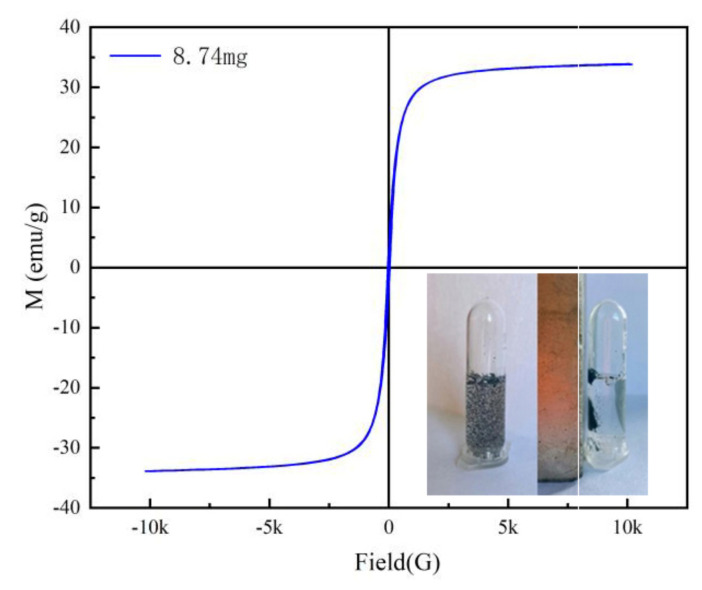
Magnetic separation and hysteresis curve of Fe_3_O_4_@rGO.

**Figure 6 toxins-16-00057-f006:**
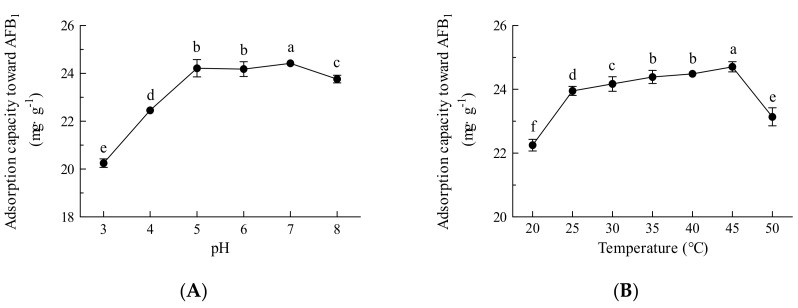
Effects of pH and temperature on the adsorption capacity of Fe_3_O_4_@rGO for AFB_1_. (**A**) Optimum pH for adsorption; (**B**) optimum temperature for adsorption. The data are shown as the means ± SDs from three biological replicates and were analyzed using Duncan’s multiple range test using the SPSS v17.0 data processing system. The different lowercase letters indicate significant differences (*p* < 0.01).

**Figure 7 toxins-16-00057-f007:**
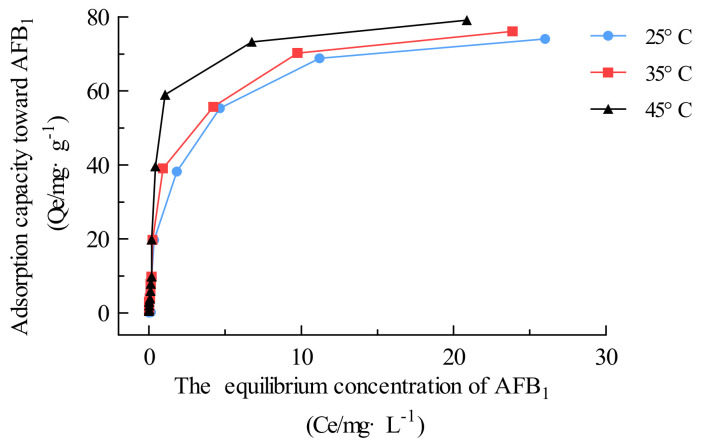
Adsorption isotherm plots and fitting curves of Fe_3_O_4_@rGO for AFB_1_ at different temperatures. Experimental conditions: initial AFB_1_ concentration, 0.02–100 mg·L^−1^; adsorbent dosage, 1 mg·mL^−1^.

**Figure 8 toxins-16-00057-f008:**
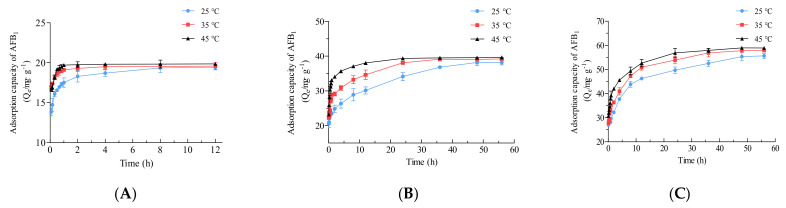
Adsorption kinetic curves of Fe_3_O_4_@rGO forAFB_1_ at different temperatures with diverse initial AFB_1_ concentrations. (**A**) 20 mg·L^−1^ initial AFB_1_ concentration; (**B**) 40 mg·L^−1^ initial AFB_1_ concentration; (**C**) 60 mg·L^−1^ initial AFB_1_ concentration.

**Table 1 toxins-16-00057-t001:** Effect of preparation conditions on the adsorption capacity of Fe_3_O_4_@rGO.

Temperature (°C)	Time (h)	Adsorption Capacity for AFB_1_ (mg·g^−1^) *
180	4	30.51 ± 0.65 ^d^
200	4	46.52 ± 0.60 ^b^
220	4	26.22 ± 0.75 ^f^
180	6	33.08 ± 0.45 ^c^
200	6	50.25 ± 0.71 ^a^
220	6	28.33 ± 0.62 ^e^
180	8	33.65 ± 0.48 ^c^
200	8	49.76 ± 0.08 ^a^
220	8	29.96 ± 0.62 ^d^
180	10	33.77 ± 0.26 ^c^
200	10	49.56 ± 0.06 ^a^
220	10	28.86 ± 0.59 ^d e^

* The data are shown as the means ± SDs from three biological replicates and were analyzed via Duncan’s multiple range tests using the SPSS v17.0 data processing system; the different lowercase letters indicate significant differences (*p* < 0.01).

**Table 2 toxins-16-00057-t002:** Isothermal equation fitting parameters of Fe_3_O_4_@ rGO for the adsorption of AFB_1_.

*T* (°C)	Langmuir Model			Freundlich Model			Temkin Model		
	*Q*max (mg·g^−1^)	*K*L (L·mg^−1^)	*R*2	*n*	*K*F (mg·g^−1^)	*R*2	*A*T (L·mg^−1^)	*B*T (J·mol^−1^)	*R*2
25	76.34	0.94	0.9970	1.74	20.63	0.9557	45.28	9.386	0.8708
35	78.74	1.02	0.9975	1.71	21.21	0.9599	51.39	9.532	0.8919
45	82.64	1.19	0.9968	0.57	28.23	0.9049	51.40	9.520	0.8696

**Table 3 toxins-16-00057-t003:** Kinetic fitting parameters of Fe_3_O_4_@rGO for AFB_1_ adsorption.

*T* (°C)	AFB_1_ *C*_0_ (mg·mL^−1^)	*Q*_e,exp_ (mg·g^−1^)	Pseudo-First-Order Kinetic Model			Pseudo-Second-Order Kinetic Model		
			*K*_1_ (min^−1^)	*Q*_e1,cal_ (mg·g^−1^)	*R*2	*K*_2_ (g·mg·min−1)	*Q*_e2,cal_ (mg·g−1)	*R*2
25	20	19.66	0.8595	12.58	0.8252	0.2257	19.80	0.9998
	40	38.17	0.2268	26.05	0.7256	0.0147	34.97	0.9858
	60	55.34	0.3723	48.91	0.9558	0.0148	53.48	0.9979
35	20	19.76	0.4093	1.75	0.8516	1.2954	19.65	1.0000
	40	39.07	0.4635	45.45	0.7455	0.0161	39.37	0.9948
	60	57.77	0.2140	49.86	0.6510	0.0107	48.78	0.9870
45	20	19.84	0.5773	1.08	0.8609	3.6288	19.84	1.0000
	40	39.58	0.8457	40.78	0.9216	0.0643	39.84	0.9998
	60	58.94	0.3019	46.10	0.9458	0.1380	54.95	0.9960

**Table 4 toxins-16-00057-t004:** Thermodynamic fitting parameters of Fe_3_O_4_@rGO for AFB_1_ adsorption.

AFB_1_ *C*_0_ (mg·g^−1^)	*T* (°C)	Δ*H*^0^ (kJ·mol^−1^)	Δ*S*^0^ (J·mol ^−1^)	Δ*G*^0^ (kJ·mol^−1^)	*R* ^2^
20	25	29.41	132.38	−10.04	0.9990
	35			−1.36	
	45			−12.68	
40	25	58.15	220.27	−7.49	0.9991
	35			−9.70	
	45			−11.90	
60	25	59.49	220.27	−6.15	0.9997
	35			−8.35	
	45			−10.56	

**Table 5 toxins-16-00057-t005:** Effects of Fe_3_O_4_@rGO dosage on the adsorption of natural active substances.

Treated Food	Food Quality	Fe_3_O_4_@rGO Dosage (mg·mL^−1^)				Activated Charcoal Dosage (mg·mL^−1^)			
		0	1	2	4	0	40	80	160
Vegetable oil	AFB_1_ adsorption rate (%)	0	21.59 ± 1.31	80.87 ± 1.96	93.72 ± 1.48	0	23.34 ± 1.37	47.30 ± 2.55	55.63 ± 0.33
	Food loss rate (%)	0	2.48 ± 0.52	3.72 ± 0.52	4.78 ± 0.94	0	12.90 ± 1.26	20.39 ± 0.39	28.54 ± 2.56
	Oryzanol content (%)	0.16 ± 0.012	0.13 ± 0.005	0.13 ± 0.002	0.13 ± 0.002	0.16 ± 0.012	0.13 ± 0.007	0.13 ± 0.005	0.13 ± 0.005
	Phosphatidecontent (mg·g^−1^)	2.84 ± 0.07	2.49 ± 0.03	2.28 ± 0.06	2.18 ± 0.03	2.84 ± 0.07	2.30 ± 0.04	2.14 ± 0.03	2.09 ± 0.03
Peanut milk	AFB_1_ adsorption rate (%)	0	25.67 ± 1.55	79.79 ± 0.60	94.44 ± 1.18	0	14.74 ± 1.74	45.43 ± 1.72	52.73 ± 0.69
	Food loss rate (%)	0	4.62 ± 0.06	6.16 ± 0.31	7.57 ± 0.26	0	12.82 ± 0.27	19.69 ± 1.81	27.80 ± 0.92
	Proanthocyadin content (μg·mL^−1^)	53.29 ± 1.22	52.73 ± 1.02	52.37 ± 0.60	52.11 ± 1.11	53.29 ± 1.22	48.41 ± 0.40	46.76 ± 0.52	46.39 ± 0.57
	Resveratrol content (μg·mL^−1^)	1.50 ± 0.03	1.43 ± 0.01	1.41 ± 0.01	1.41 ± 0.03	1.50 ± 0.03	1.38 ± 0.01	1.33 ± 0.02	1.28 ± 0.01

## Data Availability

Data is contained within the article.

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
