# Peer review of "A Magnetic Reduced Graphene Oxide Nanocomposite: Synthesis, Characterization, and Application for High-Efficiency Detoxification of Aflatoxin B1"

_toxins, 2024, doi:10.3390/toxins16010057_

Round 1

Reviewer 1 Report

Comments and Suggestions for Authors

The manuscript is well-prepared, and the experiments were performed well.

The subject of the study is needed in the field.

Suggestions:

Safety concerns about using magnetic reduced graphene oxide composite in food are to be included in the manuscript.

Highlight the limitations of the prepared composite!

How can we apply this composite in another field?

Explain the rationale for the high absorption capacity of the prepared composite.

Toxicity of the Fe3O4@rGO

mechanism involved in the adsorption process of AFB1 by Fe3O4@rGO

specific interactions between the composite material and AFB1 molecules

existing methods for AFB1 detoxification

limitations of the proposed nanocomposite, such as scalability, cost, or any specific conditions required for optimal performance

Recent references are very less (only three)

Reviewer 2 Report

Comments and Suggestions for Authors

This study deals with an important topic focusing on the synthesis, Characterization, and Application of a magnetic reduced graphene oxide nanocomposite for a high-efficient detoxification of Aflatoxin B1. The study is well designed and the results are well presented and discussed. The manuscript is well written, however, there are some corrections and suggestions that should be applied to improve the quality of the work. In particular, statistical analysis of the obtained data should be added to most of the tables and figures. In addition, some minor text corrections should be applied. All corrections and suggestions are existed in the attached version of the manuscript. So, that i recommend a major revision before the acceptance of the manuscript.   

Comments on the Quality of English Language

Reviewer 3 Report

Comments and Suggestions for Authors

The work is processed at a very good level and corresponds to the focus of your magazine, I have only slight comments about the work.

Keywords are repeated in the title.

The introduction mentions the risk of contamination of livestock feed and the risk of transmission of flatoxin to the human population. I recommend that the article also mentions the fact that the contamination of feed and flatoxins has a negative effect on the health of farm animals and give examples in pigs and ruminants.

Round 2

Reviewer 1 Report

Comments and Suggestions for Authors

Accept the manuscript.

Reviewer 2 Report

Comments and Suggestions for Authors

Almost the requested comments were covered by the authors but still minor corrections before accepting for publication. Please see the attached file.

Comments on the Quality of English Language
